# Effects of Multiple Reflow on the Formation of Primary Crystals in Sn-3.5Ag and Solder Joint Strength: Experimental and Finite Element Analysis

**DOI:** 10.3390/ma16124360

**Published:** 2023-06-13

**Authors:** Siti Farahnabilah Muhd Amli, Mohd Arif Anuar Mohd Salleh, Mohd Sharizal Abdul Aziz, Hideyuki Yasuda, Kazuhiro Nogita, Mohd Mustafa Al Bakri Abdullah, Ovidiu Nemes, Andrei Victor Sandu, Petrica Vizureanu

**Affiliations:** 1Center of Excellence Geopolymer & Green Technology (CeGeoGTech), University Malaysia Perlis (UniMAP), Taman Muhibbah, Kangar 02600, Malaysia; sitifarahnabilah@outlook.com (S.F.M.A.); mustafa_albakri@unimap.edu.my (M.M.A.B.A.); 2Faculty of Chemical Engineering Technology, University Malaysia Perlis (UniMAP), Taman Muhibbah, Kangar 02600, Malaysia; 3School of Mechanical Engineering, University Sains Malaysia, Nibong Tebal, Gelugor 14300, Malaysia; msharizal@usm.my; 4Department of Materials Science and Engineering, Kyoto University, Sakyo-ku, Kyoto 606-8501, Japan; yasuda.hideyuki.6s@kyoto-u.ac.jp; 5Nihon Superior Centre for the Manufacture of Electronic Materials (NS CMEM), School of Mechanical and Mining Engineering, The University of Queensland, Brisbane, QLD 4072, Australia; k.nogita@uq.edu.au; 6Department of Environmental Engineering and Sustainable Development Entrepreneurship, Faculty of Materials and Environmental Engineering, Technical University of Cluj-Napoca, B-dul Muncii 103-105, 400641 Cluj-Napoca, Romania; 7Faculty of Materials Science and Engineering, Gheorghe Asachi Technical University of Iasi, Blvd. D. Mangeron 71, 700050 Iasi, Romania; sav@tuiasi.ro (A.V.S.); peviz@tuiasi.ro (P.V.); 8Romanian Inventors Forum, Str. Sf. P. Movila 3, 700089 Iasi, Romania; 9Technical Sciences Academy of Romania, Dacia Blvd 26, 030167 Bucharest, Romania

**Keywords:** soldering, primary intermetallics, synchrotron, multiple reflow, finite element

## Abstract

The growth and formation of primary intermetallics formed in Sn-3.5Ag soldered on copper organic solderability preservative (Cu-OSP) and electroless nickel immersion gold (ENIG) surface finish after multiple reflows were systematically investigated. Real-time synchrotron imaging was used to investigate the microstructure, focusing on the in situ growth behavior of primary intermetallics during the solid–liquid–solid interactions. The high-speed shear test was conducted to observe the correlation of microstructure formation to the solder joint strength. Subsequently, the experimental results were correlated with the numerical Finite Element (FE) modeling using ANSYS software to investigate the effects of primary intermetallics on the reliability of solder joints. In the Sn-3.5Ag/Cu-OSP solder joint, the well-known Cu_6_Sn_5_ interfacial intermetallic compounds (IMCs) layer was observed in each reflow, where the thickness of the IMC layer increases with an increasing number of reflows due to the Cu diffusion from the substrate. Meanwhile, for the Sn-3.5Ag/ENIG solder joints, the Ni_3_Sn_4_ interfacial IMC layer was formed first, followed by the (Cu, Ni)_6_Sn_5_ IMC layer, where the formation was detected after five cycles of reflow. The results obtained from real-time imaging prove that the Ni layer from the ENIG surface finish possessed an effective barrier to suppress and control the Cu dissolution from the substrates, as there is no sizeable primary phase observed up to four cycles of reflow. Thus, this resulted in a thinner IMC layer and smaller primary intermetallics, producing a stronger solder joint for Sn-3.5Ag/ENIG even after the repeated reflow process relative to the Sn-3.5Ag/Cu-OSP joints.

## 1. Introduction

In electronic packaging, solder joints serve electrical and mechanical support between the solder and substrates. It is well known that the intermetallic compound (IMC) will form after the solder reacts with the substrate during the soldering process. The presence of the IMCs between solder joints is an essential requirement of good metallurgical bonding and excellent joint reliability. Until now, the most critical factor in the solder interconnects is the formation of IMCs at the interfacial layer and in the solder matrix. Excessive growth of the IMC layer could degrade the reliability of the solder joints due to its brittle nature. In the three-dimensional (3D) electronic packaging technology, most component electronic products may experience more than one cycle of reflow (multiple reflows), causing more interactions between the solder and substrate, forming large and coarse brittle IMCs. Therefore, the solid–liquid–solder interaction between the solder and substrate is essential to investigate since it could affect the solder joint’s reliability.

In previous studies, significant efforts have been made to investigate the effect of surface finishes on the formation of intermetallics, intensively during the reflow soldering and thermal aging [1,2]. Unfortunately, only a few studies were conducted on the influence of surface finish during multiple reflows. For instance, Zhong et al. analyzed the effect of solder joint reliability after multiple reflows of tin–lead (SnPb) and tin–silver–copper (Sn-Ag-Cu) with the addition of indium (In) on the Au/Ni substrate [3]. They discovered that the Sn-Ag-Cu-In strength slightly decreased after the fourth reflow, relative to the (Sn-Ag-Cu) SAC and SnPb, due to the thermal degradation in solder/IMC and IMC/Ni interfaces. On the other hand, the evolution of the interfacial reaction between the SAC and SnPb solder joints with organic solderability preservative (OSP) and electrolytic nickel/gold (Ni/Au) substrate during multiple reflows was investigated by Liu et al. [4]. They established that the thickness of the interfacial IMC layer increases more rapidly in OSP solder joints with the increasing number of reflows due to the faster Cu diffusion. Moreover, Chen et al. discovered that the phosphorus (P) content of the electroless Ni-P layer also influences the formation of the interfacial IMC layer and solder joint strength of the Cu/electroless Ni-P/Sn-3.5Ag after multiple reflows [5]. The formation of Ni_3_Sn_4_ in the Cu/Ni-P/Sn-3.5Ag solder joint is mostly spalled into the solder area near the interfacial layer. However, the loose spallation of Ni_3_Sn_4_ IMC was not found to influence the reliability of the solder joints.

Many available works reported the effect of different surface finishes detailing the formation of the interfacial IMC layer after multiple reflows and the influence on the solder joint reliability. However, the study of the formation of primary IMC on the solder joint strength after multiple reflow cycles remained insufficient. In the previous study we conducted, the formation of primary Cu_6_Sn_5_ intermetallics and the strength of the solder joint based on the two types of surface finish, which are copper organic solderability preservative (Cu-OSP) and electroless nickel immersion gold (ENIG), were examined [6]. In this paper, the effect of Sn-3.5Ag soldered on two types of surface finish, which are Cu-OSP and ENIG surface finish substrate, during multiple cases of reflow soldering were explored. The evolution of primary IMCs during six times of repeated reflow cycles was evaluated. Furthermore, the influence of primary IMCs on the solder joints subjected to the multiple reflows was carried out using the experimental high-speed shear test and the finite element analysis (FEA).

## 2. Experimental Procedures

### 2.1. Sample Preparation

In this research, solder ingots of Sn-3.5Ag were cast utilizing a top-loading furnace with casting temperature of 350 °C for 30 min. Then, molten solder was poured onto stainless steel plates and rolled into foils with the thickness of 70 μm. The thin foils were then punched with a puncher of 2.5 mm to produce solder spheres. The rosin mildly activated (RMA) was applied on the punched solder and placed on a Pyrex sheet. Correspondingly, it was heated in an oven at a temperature of 250 °C for 60 s and formed into solder balls with a diameter of ~900 μm. The solder balls were cleaned and sieved to ensure the uniformity of the sample sizes. In order to form solder joints, the solder balls of Sn-3.5Ag were arranged on the Cu-OSP and ENIG substrates and a small amount of RMA flux were applied and soldered in a desktop reflow oven at 250 °C for 60 s. The OSP and ENIG layer thickness on the Cu substrate was ~0.2–0.6 μm and ~5.5–7.7 μm, respectively, with a ~900 μm pad opening, as depicted in Figure 1a,b. The remaining flux at the samples were cleaned thoroughly using acetone before proceeding to the test.

### 2.2. Real-Time In Situ Synchrotron Imaging

The synchrotron radiography imaging was conducted at beamline BL20XU at Spring-8 synchrotron in Hyogo, Japan, using the solidification observation setup described previously [7]. The purpose of the in situ synchrotron technique was to investigate the formation and evolution of the primary intermetallics during the process of soldering. The samples were prepared by vertically placing thin 100 μm sheets of Sn-3.5Ag solder on a thin 100 μm Cu-OSP and ENIG substrates. Subsequently, the samples were then placed between a 100 μm polytetrafluoroethylene (PTFE) sheet with a vent to enable flux outgassing and held between two quartz plates. Consequently, the samples were set to be heated from room temperature up to 250 °C at 0.33 °C/s and held for 30 s at 250 °C before being cooled down at 0.33 °C/s; this was repeated for 6 cycles. All the images were captured with an exposure time of 1 s per 10 frames (as reflowed) and 1 s per frame (subsequent reflow). The field of view was 1024 × 1024 µm^2^ with resolution ratio of 0.5 µm per pixel.

### 2.3. Microstructure Characterization

After soldering and subsequent multiple reflows, the solder joints samples (including the post-experiment in situ samples) were embedded in epoxy resin mixed with epoxy hardener and mechanically grounded with the silicon carbide (SiC) paper before proceeding to the polished steps using the colloidal silica suspension and oxide polishing suspensions (OPS). Detailed cross-sectional studies were conducted using a scanning electron microscope (SEM) of JEOL JSM 6460LA with an accelerating voltage of 25 kV equipped with an energy dispersive spectroscopy (EDS). In addition, the thickness of the interfacial IMC was measured by measuring 10 different samples with various interfacial positions. The measurement was performed by dividing the area of IMC over the length utilizing ImageJ software.

### 2.4. Solder Joint Strength

To study the strength of the solder joint of Sn-3.5Ag/Cu-OSP and Sn-3.5Ag/ENIG after being reflowed for 6 cycles, the samples were mechanically tested using a Dage 4000 high-speed bond tester at the Nihon Superior Research and Development (R&D) in Osaka, Japan. A lower speed of 100 mm/s and a high speed with 2000 mm/s shear speed with a 90 μm shear height were used during the testing. For a more accurate observation of the solder joint strength, 16 samples were tested for each condition. After the shear test, the acetone was used to clean and rinse the samples. Then, the fracture behavior of the solder joint was observed using a JEOL JSM 6460LA SEM (Tokyo, Japan). 

### 2.5. Finite Element Analysis

In the present study, a 3D FE simulation was utilized to predict the effect of the solder joint strength by focusing on the primary intermetallics in the Sn-3.5Ag solder joint. A commercial FE using an ANSYS Release 19 was employed to perform the computational analysis. The solder joint geometric model was constructed using Solidworks software, followed by the actual dimension from the experiment, as in Figure 2a, and consists of the solder ball and primary intermetallics. Figure 2b,c illustrate the components of the FE simulation, including a reflowed solder ball, Cu_6_Sn_5_ intermetallics, shear tools, and a 3D FEA mesh model, respectively. The FE was also utilized to study the effect of different distributions on the solder joints, such as the primary intermetallics distributed homogeneously in the solder bulk, centered and peripheral, with a primary length of 50–250 µm, as depicted in Figure 3. Note that the shear tool was considered a rigid body that moves horizontally from left to right, while the bottom sides of the solder ball and copper pad are fixed to remove the rigid body motion. From the model presented in this study, there were 75,657 nodes and 49,322 elements. All the materials (except the solder ball) were considered linear–elastic, while the solder ball was set as elastic–viscoplastic. The properties of the constituent components are presented in Table 1 and Table 2, and were obtained from several studies in the literature. The creep behavior is governed by the equation provided below to determine the exact deformation behavior of the solder (1) [8]:(1)$$\frac{d\varepsilon }{dt}=C1{\left[\sin h\left(C2\sigma \right)\right]}^{C3}\exp\left(-\frac{C4}{T}\right).$$

## 3. Results and Discussion

### 3.1. *In Situ* Imaging of Primary Intermetallic Growth during Multiple Reflows

In situ synchrotron X-ray imaging with a peak temperature of 250 °C was used to observe the nucleation and growth of the primary crystals in the solder of Sn-3.5Ag soldered on Cu-OSP and ENIG surface finish during multiple reflows. Figure 4 and Figure 6 demonstrate the solidification of the Cu_6_Sn_5_ and Ni_3_Sn_4_, and the (Cu, Ni)_6_Sn_5_ primary crystals for Sn-3.5Ag/Cu-OSP and Sn-3.5Ag/ENIG solder joints for each reflow cycle. The dark rods in the solder ball are the primary IMCs, the grey part is the Sn liquid solder, and the brighter colors are the shallow bubbles at the central solder. Apart from that, the interface between the solder Sn-3.5Ag and substrate are flux voids. The formation of bubbles was due to the flux outgassing process, where some gases are trapped at the solder interface during the soldering process. The size of the bubble increased gradually with the increasing temperature. As can be observed from Figure 4a–d, there are “multi-bubbles” located at the interface during the whole soldering process that remained even after the fourth time reflow with increased volume. The “multi-bubbles” were obtained from an interaction of small bubbles with each other and expanded to a larger bubble volume. After reaching a specific time, some bubbles, including the “multi-bubbles,” disappeared due to the high-pressure content. Nevertheless, these will not be discussed further in this study.

As depicted in Figure 4, it can be seen that the Cu_6_Sn_5_ primary intermetallics formed in various morphologies that consist of a mixture of a hexagonal rod, hollow faceted hexagonal rods, and “in-plane” branched form. Several studies reported that the different morphologies of Cu_6_Sn_5_ could be influenced by different factors, including the Cu content, the undercooling, and the cooling rate [13,14,15]. In this research, it is observed that the formation of primary Cu_6_Sn_5_ crystals in a joint of Sn-3.5Ag/Cu-OSP can be categorized into three solidified locations based on reflow cycles: (i) first reflow, (ii) from second to fourth reflow, and (iii) from fifth to sixth cycle; these will be discussed later. The first category, shown in Figure 4a, represents the solidification of the as-reflowed joints. Note that many primary Cu_6_Sn_5_ intermetallics with the hexagonal rod shape began nucleating and growing at the interface area and large faceted hexagonal rods with hollow areas marked as A, B, and C. During the early nucleation, the Cu_6_Sn_5_ crystal (A, B, and C) grew as filled rods and, after a few seconds, the hollowness started to widen perpendicularly to the growth tip along the rods. Additionally, the “in-plane” branched type of primary Cu_6_Sn_5_ nucleated in the Sn-3.5Ag/Cu-OSP was observed, marked D, E, F, G, and H. The growth kinetics and the apparent number of primary Cu_6_Sn_5_ in the Sn-3.5Ag/Cu-OSP were measured and plotted in a graph shown in Figure 5.

After the first reflow cooled to ~173 °C at a rate of 0.33 °C/s, the solder was reflowed again for up to six cycles with the similar solder joint and temperature profile used at the first cycle. For the second category, it can be seen that the solidification of the primary Cu_6_Sn_5_ in the Sn-3.5Ag/Cu-OSP joint after the second reflow has a preferred nucleation point up to the fourth reflow cycle, where the primary intermetallic formed was quite similar, as presented in Figure 4b–d. However, as can be seen, the growth location of Cu_6_Sn_5_ primary during solidification for the second cycle was different from the as-reflowed joints in Figure 4a. Similar to the as-reflowed joints, the morphology of the Cu_6_Sn_5_ that formed after the second reflow is in the mixture of hexagonal rods, faceted hollow rods, and “in-plane” branched but is different in the number of crystals formed. It can be concluded that the number of primary Cu_6_Sn_5_ crystals increased considerably with the increasing reflow cycle due to the increment of Cu diffused from the substrate during the soldering process, as plotted in Figure 5. Additionally, increasing the number of reflows, the average growth rate of the formation of Cu_6_Sn_5_ primary found became slower from 16.67 µm/s (first cycle) to 0.86 µm/s (sixth cycle) due to the diffusion barrier caused by the formation of IMC denser at the interface area and the saturation of the solder. Moreover, some transition of the primary Cu_6_Sn_5_ crystals’ morphologies occurs from the hexagonal rods to the in-plane branched due to the increased Cu diffused after multiple reflow cycles.

The third category refers to the fifth and sixth cycles of the Sn-3.5Ag/Cu-OSP solder joint, with a similar pattern in the nucleation location and growth of primary Cu_6_Sn_5_ that formed as displayed in Figure 4e,f. During the fifth reflow process, a continuous layer of multi-bubbles suddenly burst and disappeared due to the higher gas pressure inside the bubbles [16]. Henceforth, it is seen that the Cu_6_Sn_5_ primary began nucleating and growing from the interface area, as shown in Figure 4e. It may be inferred that the formation of bubbles and interfacial voids at the interface can affect the growth of the IMC. As studied by Kunwar et al., the formation of bubbles at the interface could hinder the diffusion of Cu at the area of the bubbles and the resulting poor strength of the solder joints [17]. As per Figure 4e,f, most primary Cu_6_Sn_5_ formed are the “in-plane” branched type, and some of the primary crystals that start nucleates near the interface area are formed in the faceted hollow rods. Other than that, the development of the “in-plane” branched Cu_6_Sn_5_ in Sn-3.5Ag/Cu-OSP after the fifth cycle was added to by the continuous Cu diffusion from the substrate into the solder during multiple reflows, resulting in a higher concentration of Cu in the matrix of the solder. During the sixth reflow, the solder and primary IMC only partially melted due to the saturation limit. As can be observed, no new primary IMC can be formed since the β-Sn nucleates faster due to the saturation limit of the solder. Only a few pre-existing primary Cu_6_Sn_5_ partially melt during the heating and continue to grow until they are solidified.

Figure 6 illustrates a synchrotron radiograph of the solder joint solidification of Sn-3.5Ag/ENIG after multiple reflows six times. As seen in Figure 6, there is a trapped interfacial void form at the interface; it continually expanded and remained until the sixth cycle. The increase in the radius void can be explained due to the heating process where the solder joint is subjected to multiple reflows. After being reflowed, a small amount of Ni_3_Sn_4_ was formed at the interface area because the Ni atoms diffused from the substrate and reacted with the Sn solder alloy. Apart from that, the formation of Ni_3_Sn_4_ was observed to be concentrated at the bottom area near the solder joint and remained after prolonged reflow of six numbers. No visible significant primary phases were observed during the multiple reflows up to the third cycle. However, the nucleation and growth of the small size of the primary (Cu, Ni)_6_Sn_5_ were observed at the bulk solder area after five reflows. This was found due to the limited amount of Ni content in a thin layer of ENIG surface finish (~5–7 µm), where some of the Ni atoms were already consumed at the previous reflow cycles to form Ni_3_Sn_4_ at the interfacial. Moreover, the (Cu, Ni)_6_Sn_5_ that formed in the Sn-3.5Ag/ENIG solder joint could be due to the Cu dissolution from the substrate during the process of soldering after the fifth reflow cycle. 

It can be confirmed via the EDS analysis of the synchrotron sample after the sixth reflow, as shown in Figure 7b; small (Cu, Ni)_6_Sn_5_ intermetallics started forming at solidified at the fifth cycle. Additionally, by using the EDS analysis as per Figure 7b at point 3, a small Ag_3_Sn was detected at the interfacial area after the third reflows, whereas, in the Sn-3.5Ag/Cu-OSP samples, the growth rate of (Cu, Ni)_6_Sn_5_ also decelerated from 1.82 µm/s at the fifth cycle to 0.48 µm/s due to the concentration at the interface area. The slower growth rate of IMC with respect to the multiple reflows could be attributed to the changing diffusion path of the atoms during the solid–liquid–solid interaction. Additionally, the initial IMC layers at the interface formed after the as-reflowed joint might retard the dissolution during the subsequent reflow.

### 3.2. Interfacial Intermetallic Layer Formation and Growth after Multiple 

A continuous intermetallic layer can provide reliable interconnection bonding between the solder and substrate with an appropriate thickness in the solder joint. However, excessive growth of the IMC’s layer will degrade the solder joint reliability. Figure 8 and Figure 9 depict the SEM cross-sectional images of the solder joints in Sn-3.5Ag/Cu-OSP and Sn-3.5Ag/ENIG focusing on the interfacial IMC after multiple reflows up to six cycles, respectively. These samples were prepared separately from the in situ synchrotron imaging experiments. As shown in Figure 8, the Cu-Sn IMC layer is generated at an interface between the solder and substrate for Sn-3.5Ag/Cu-OSP solder joint. Using the EDS, the Cu_6_Sn_5_ phase was detected where it was composed of Cu 53.61 at.% and Sn 43.46 at.%. The Cu_6_Sn_5_ layer forms due to the heterogenous nucleation and the Cu diffusion from the substrate into the solder at the interface reaction between the Sn and Cu. The Ag_3_Sn also formed in small sizes that were finely dispersed into the eutectic region. At the as-reflowed joint of the Sn-3.5Ag/Cu-OSP solder joint, a scallop-liked Cu_6_Sn_5_ IMC layer was observed at the interface. It is worth noting that the scallop-liked Cu_6_Sn_5_ grew and became thicker, transforming into planar-liked morphology after a prolonged reflow of sixth cycles. The changes could be attributed to the higher diffusion paths at the channels between the scallops that allow the Cu from the substrate to diffuse faster through it and also owing to the ripe flux of the Cu_6_Sn_5_ layer growth [18]. Additionally, fewer cracks also occur in the Cu_6_Sn_5_ layer parallel to the substrate after the fifth reflow cycle in Figure 8e. These cracks’ formation is caused by the stress generated in the IMCs during the multiple reflow process.

Figure 9a–f represents the Sn-3.5Ag/ENIG solder joint’s micrograph after being reflowed six times. It can be observed that, after the Sn-3.5Ag solder reacts with the Ni-containing surface finish, a loose-like thin Ni_3_Sn_4_ intermetallic layer was formed at the interface. It also can be seen that there is a dark Ni-P-rich layer present underneath the Ni_3_Sn_4_, marked as a red dash line in Figure 9a. With the help of EDS analysis, it was confirmed that the interfacial microstructure in Sn-3.5Ag/ENIG joints consists of Ni_3_Sn_4_ and a Ni-P rich IMC layer composed of Ni (43.28 at.%), Sn (56.72 at.%), Ni (81.40 at.%), and P (11.07 at.%), respectively. As stated by Kumar et al., the P element was produced during the electroless Ni plating process and then reacted with the Ni layer substrate to form the Ni_3_Sn_4_ [19]. This observation aligns with previous studies that pointed out that the Ni_3_Sn_4_ formed at the interface between Sn-3.5Ag and the Ni substrate [5]. Even though the Ni_3_Sn_4_ IMC formed as a spallation on the Sn-3.5Ag/ENIG solder joints, it was discovered to have no effect on the strength of the solder joints. Note that a thin layer of (gold) Au appeared to have fully dissolved into the solder matrix after being reflowed. Nevertheless, no Au element detected in the interface area might be due to the detection limit of EDS [20]. Additionally, no large Cu-Sn-Ni IMC was found in the Sn-3.5Ag/ENIG solder joint due to the Ni layer’s capability to suppress the Cu dissolution from the substrate up to the third cycle reflow. However, there was a Cu-Sn-Ni IMC formation detected at the sixth reflow cycles, as found in Figure 7b point 4 with the IMC composed of Ni (11.23 at.%), Sn (74.92 at.%), and Cu (13.64 at.%), since the dissolution of Cu can occur after the fourth reflow cycle. Similar to the Sn-3.5Ag/Cu-OSP joints, as the number of reflow cycles increased, the size and thickness of the Ni_3_Sn_4_ interface layer increase as well, as depicted in Figure 9f.

The quantitative measurement of the thickness of each solder joint was conducted after the sixth reflow cycle by dividing the total area of the IMC layer and the total length of the IMC. The mean thickness layer for each cycle was plotted in Figure 10a to compare the different thickness growths over multiple reflow cycles on both solder joints. The results indicate that the mean thickness of the Cu_6_Sn_5_ layer at the Sn-3.5Ag/Cu-OSP interface increased from ~4.98 μm after the first reflow to a maximum of ~11.28 after the sixth cycle. For the Sn-3.5Ag/ENIG solder joints, the mean thickness of the Ni_3_Sn_4_ layer grew from ~2.03 μm after being reflowed to ~4.25 μm after the sixth reflow. Both the thicknesses of Cu_6_Sn_5_ and Ni_3_Sn_4_ IMC’s layers increased linearly as the joints were subjected to multiple reflows. However, the Cu_6_Sn_5_ IMC layer on Sn-3.5Ag/Cu-OSP was ~3 times thicker relative to the Ni_3_Sn_4_. This can be attributed to the rapid growth of the interfacial layer from the dissolution/diffusion of Cu from the substrate.

Additionally, the thin layer of Ni_3_Sn_4_ in the Sn-3.5Ag/ENIG solder joint that appears after performing multiple reflows could be due to the Ni layer’s effectiveness in acting as a diffusion barrier. Besides that, the formation of this thin layer was due to the formation of a pre-existing IMC layer at the initial reflow that helps to inhibit the diffusion between Cu from the substrate and solder. These results agreed with the study by Zhong et al. [3]. According to the literature, a Cu_3_Sn IMC layer may be present at the interface between the Cu_6_Sn_5_ and Cu substrate after exposure to high-temperature aging. Nevertheless, there are no Cu_3_Sn phases detected on the Sn-3.5Ag/Cu-OSP interface due to the limitation of the characterization used.

It is known that the kinetic growth of the IMC layer can be either diffusion controlled or interfacial reaction controlled. The relationship between the thickness of the interfacial layer with respect to multiple reflows can be expressed according to the power–law equation as follows: *W* = *kt^n^*,(2)
where *W* is the average thickness of the interfacial layer, *k* is the growth rate constant, *t* is the reflow time above the liquidus temperature during reflow soldering, and *n* is the time exponent. By plotting the graph of the interfacial thickness against the reflow time (Figure 10b) and against the square root of the reflow time (Figure 10c), both the *k* and *n* values are obtained from the slope of the linear regression curve, respectively. The IMC growth mechanism is controlled by grain boundary diffusion if the *n* is 0.33. When the time exponent time *n* is 0.5 or 1, the growth mechanism of the interfacial layer is controlled by a bulk diffusion-controlled or an interface reaction rate-controlled process, respectively [4]. Note that the value of the time exponent *n* in this study for Cu_6_Sn_5_ in Sn-3.5Ag/Cu-OSP and Ni_3_Sn_4_ in Sn-3.5Ag/ENIG joint was 0.47 and 0.42, respectively. The exponent time of the Cu_6_Sn_5_ and Ni_4_Sn_4_ values obtained was close to 0.5, indicating that a bulk diffusion-controlled mechanism governed the growth of the interfacial IMC layers. Thus, increasing the IMC layer during multiple reflows followed the square root time law, where *W = kt*^0.5^ for both solder joints. Based on the *k* value, the results present that the Sn-3.5Ag reflowed on the Cu-OSP substrate had a slightly higher growth rate constant relative to the ENIG with values of 0.58 and 0.21 µm/s, respectively. Therefore, it can be concluded that the rapid growth of the interfacial layer occurs at the Cu interface linearly with the square root of the reflow time. Additionally, it can be inferred that the Ni layer from the ENIG surface finish can be regarded as an effective diffusion barrier that hindered the formation of Cu_6_Sn_5_ intermetallics by suppressing the Cu dissolution from the substrate during the soldering process. 

### 3.3. Solder Joint Strength

The strength of the Sn-3.5Ag/Cu-OSP and Sn-3.5Ag/ENIG solder joints after six reflows was evaluated at shear speeds of 100 mm/s and 2000 mm/s. Lower shear speeds were used to determine the strength of the solder joint at the solder matrix area, while higher shear speeds were used to determine the strength near the/at the interfacial layer of the joint. Figure 11a represents the solder joint strength of Sn-3.5Ag reflowed on Cu-OSP and ENIG after the first, third, and sixth reflow cycles at 100 mm/s shear speeds. The result revealed that the Sn-3.5Ag/ENIG has a higher average shear strength at first reflowed with 57 N compared to the Sn-3.5Ag/Cu-OSP joint with an average shear strength of 40 N. When increasing the number of reflow cycles up to the sixth cycle, the Sn-3.5Ag/ENIG solder joint was observed to have slightly decreased its strength by 6% to 50 N, while, for the Sn-3.5Ag/Cu-OSP joints, this decreased by 20–35% to 25 N. Here, the joint strength decreased as the number of reflow cycles increased due to the increasing number of large brittle Cu_6_Sn_5_ primary intermetallics. 

Figure 11b illustrates the average shear strength of Sn-3.5Ag/Cu-OSP and Sn-3.5Ag/ENIG solder joints when tested at a 2000 mm/s shear speed for the first, third, and sixth reflow cycles. Similar to the solder joint strength tested at 100 mm/s shear speed, the Sn-3.5Ag/ENIG solder joints showed a higher average shear strength than Sn-3.5Ag/Cu-OSP at a 2000 mm/s shear speed. However, the results indicate that the shear strength of Sn-3.5Ag/ENIG slightly decreased by ~8% from 66 N to 56 N, while, for Sn-3.5Ag/Cu-OSP, the joint strength decreased by ~16% from 45 N to 33 N after multiple reflows. The significant decrement of Sn-3.5Ag/Cu-OSP solder joint strength after shear testing at 2000 mm/s could be attributed to excessive Cu dissolution during multiple reflows. Therefore, this resulted in a thicker formation of the Cu_6_Sn_5_ intermetallic layer at the interfacial area and also formed large brittle Cu_6_Sn_5_ primary crystals at the solder bulk area relative to the Sn-3.5Ag/ENIG joints. 

After the shear tests, the samples were inspected via SEM analysis to clarify each sheared solder joint’s failure modes. Figure 12 demonstrates the fracture surfaces mode analysis of the Sn-3.5Ag/Cu-OSP and Sn-3.5Ag/ENIG solder joints after the shear test at 100 mm/s and 2000 mm/s shear speeds. Previous researchers reported that the crack location would generally form at the bulk solder area when sheared at the low shear speed [21,22,23,24]. At the same time, the crack location for higher shear speed will form near/at the interfacial IMC area. As depicted in Figure 12a–f, it can be observed that all multiple reflow samples for both solder joints crack at the bulk solder after shear tests at 100 mm/s. From the fracture modes result, it was observed that both solder joints were in a quasi-ductile mode where more than 50% of the solder residue is left after shearing at 100 mm/s. Meanwhile, Figure 12g–l shows the crack along the interface area for solder joints when tested with a shear speed of 2000 mm/s. However, after multiple reflows, the fracture surface for both solder joints revealed a quasi-brittle mode that was significantly affected by the thickness of the IMC layer. Here, the exposed brittle IMC area was more dominant than the retained solder area.

### 3.4. Finite Element Analysis

The current study chose the shear speed of 100 mm/s for the shear test condition in Sn-3.5Ag/Cu-OSP solder joint as the base case. The effect of Cu_6_Sn_5_ primary crystal on the shear strength of the solder joints was determined. Figure 13a depicts the shear force–displacement curves obtained via computational analysis, which were compared to the experimental data of the as-reflowed Sn-3.5Ag/Cu-OSP solder joints after the shear test. From Figure 13a, the patterns of the curves from the FEA agree with the experimental data for the solder joints after being reflowed. The length of the primary crystal for FE modeling was constructed, followed by the experimental, in the range of 120–750 µm. Furthermore, the valid model compares the FEA with different sizes, amounts, and distributions of Cu_6_Sn_5_ primary intermetallics. Figure 13b–d presents the contours of the equivalent von Mises stress, total deformation, and the equivalent elastic strain distribution of the solder joints in FEA. Note that the force (shear tools) was set to move from the right to the left sides in the *x*-axis direction. The high total deformation in the solder joint was discovered to be at the tip of the solder ball after the strike with the shear ram. Additionally, the maximum contour of the von Mises stress and equivalent strain distribution occurred close to the junction of the solder ball as per Figure 13b–d. Between Figure 12a and Figure 13b–d, the computational analysis result correlates well with the experimental solder joint, where both failure modes occurred near the interface. Even though the curve trend of the results obtained is similar, the result, such as the shear force values, is expected to vary from the simulation and experiment since many factors could influence the data accuracy. Other than that, the experimental error during sample preparation and some experimental factors are not included in the computational simulation. Nevertheless, the experimental results and the computational modeling under increasing the number of primary intermetallics show a similar decrease in the solder joint strength trends. Consequently, studying the effects of primary intermetallics using FEA is reliable.

The strength of the solder joint can be affected by many aspects, including forming primary intermetallics. As reported above, the formation of the primary will increase in numbers and sizes in the solder joints after performing the multiple reflows, resulting in the poor strength of the solder joints. Using the valid model in the current study (Figure 13), the effects of the primary intermetallic were further investigated by manipulating the sizes, numbers, and distribution of the solder joints. Figure 14 illustrates the shear force–displacement curves obtained from the simulation, where 16 primary Cu_6_Sn_5_ (followed by the validation model) with different sizes were constructed. The length of the small primary in Figure 14a ranges between 50 and 120 µm, while the large primary in Figure 14b is 130–450 µm. The finding shows that the smaller-sized Cu_6_Sn_5_ crystal that formed in the bulk Sn-3.5Ag solder joints slightly increased the solder joint strength compared to the larger sizes due to its brittle nature. More shear forces are required for a smaller size than the larger size of Cu_6_Sn_5_ crystal (Figure 14c). These results aligned with our previous work, where the intermetallics formed in the SAC305 reflowed on the immersion tin (ImSn) were smaller. Note that they resulted in a higher solder joint strength than the immersion silver (ImAg) surface finish [14].

Moreover, the numbers of primary intermetallic effects in the solder joints were also conducted using FEA, as shown in Figure 15a. The primary crystals were designed in a 3D finite model using Solidworks, ranging between 130 µm and 750 µm, for 21 and 26 primaries. The results show that increased primary numbers resulted in a lower solder joint due to its brittleness. The total volume fraction was changed relatively when the Cu6Sn5 particles changed. Correspondingly, the distribution effect for each number from the data in Figure 15a was investigated, and the results are presented in Figure 15b–d. The distributions of the primary crystals were constructed into three categories: homogeneously, centerally, and peripherally distributed, as illustrated in Figure 3. A similar length of the primary crystals was used to study the effect of the distributions, as in Figure 15a. Figure 15b–d shows the shear force–displacement curves for the different distributions obtained from the simulation. It can be seen from the curves graph simulations that the distribution of the Cu_6_Sn_5_ intermetallics formed in the solder joints slightly affected the strength of the solder joints. In this experimental and modeling simulation work, it can be concluded that the solder joint’s strength depended on the sizes and number of primary intermetallics formed. At the same time, the distributions of primary crystals did not significantly affect the solder joint’s strength.

## 4. Conclusions

The growth of primary intermetallics formed in Sn-3.5Ag/Cu-OSP and Sn-3.5Ag/ENIG solder joints after six reflows were investigated. The results of this study can be summarized as follows:
The real-time in situ synchrotron imaging proves that the Cu dissolution strongly influenced the formation of primary crystals in the hypo-eutectic solder (Sn-3.5Ag) from the substrate during the multiple reflow process. There are numerous significant formations of primary Cu_6_Sn_5_ observed in the Sn-3.5Ag/Cu-OSP solder joints even after being reflowed, but not in the Sn-3.5Ag/ENIG solder joint. This is because the Ni layer from ENIG acted as an effective diffusion barrier and suppressed the Cu dissolution from the substrate up to the fourth reflow cycle.The slow growth rates of the primary (Cu, Ni)_6_Sn_5_ and interfacial IMC layer in the Sn-3.5Ag/ENIG solder joints were attributed to the ENIG-prevented Cu dissolution during the solid–liquid–solid interaction, resulting in a smaller and thinner interfacial Ni_3_Sn_4_ layer compared to the Cu_6_Sn_5_ layer on the Sn-3.5Ag/Cu-OSP.The large and thicker formation of Cu_6_Sn_5_ intermetallics present in the Sn-3.5Ag/Cu-OSP joints decreased the solder joint strength at 100 and 2000 mm/s shear speeds relative to the Sn-3.5Ag/ENIG solder joints.By performing the shear tests utilizing computational simulation using the FEA, the results of the failure mechanism and prediction of the solder joint’s strength with different numbers, sizes, and distributions of primary Cu_6_Sn_5_ intermetallics can be obtained. As a result, the solder joint strength depends on the size and number of intermetallics formed in the bulk solder rather than the intermetallic distribution.


## Figures and Tables

**Figure 1 materials-16-04360-f001:**
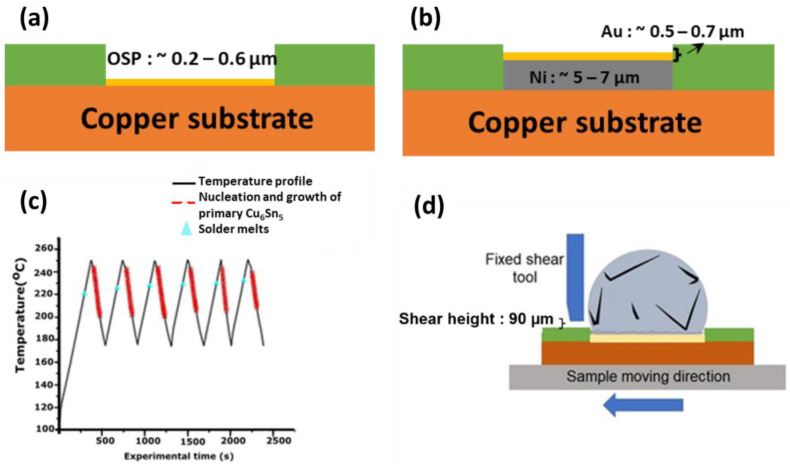
Schematic diagram of the (**a**) organic soldering preservative (OPS) and (**b**) electroless nickel immersion gold (ENIG) surface finish; (**c**) soldering temperature profile during multiple reflows of Sn-3.5Ag and (**d**) schematic diagram of the solder joint high-speed shear test.

**Figure 2 materials-16-04360-f002:**
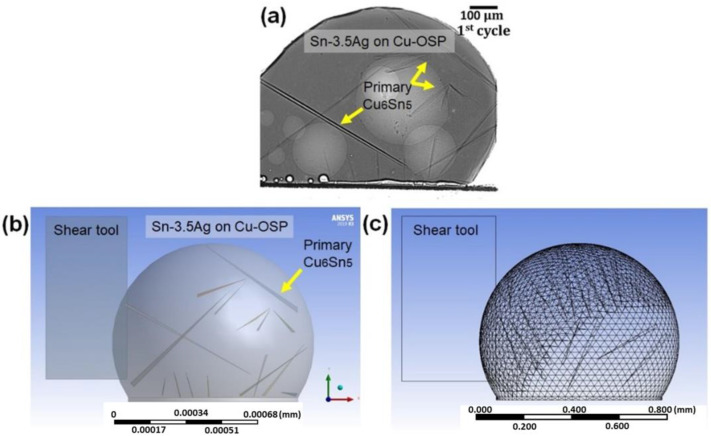
The solder joint geometric model: (**a**) Cross-sectioned view of as-reflowed Sn-3.5Ag solder joints, (**b**) Finite element model for solder ball shear tests consisting of 16 primary crystals, and (**c**) 3D FEA mesh model.

**Figure 3 materials-16-04360-f003:**
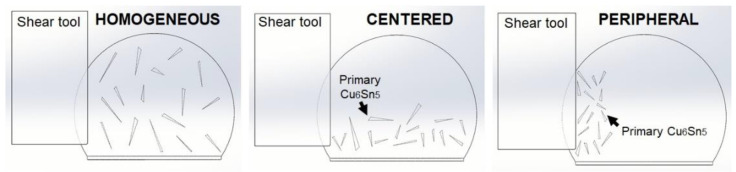
Schematic diagram of 3D finite element model with different primary intermetallics distribution.

**Figure 4 materials-16-04360-f004:**
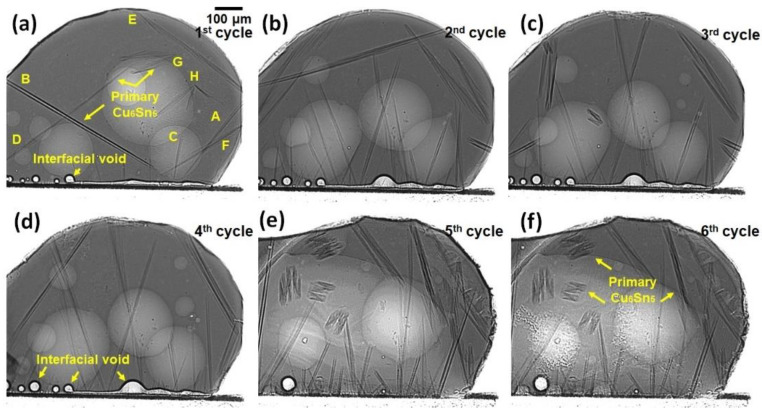
Synchrotron radiation real-time images observations of primary Cu_6_Sn_5_ formation during multiple reflows of Sn-3.5Ag on Cu-OSP surface finish: (**a**) first, (**b**) second, (**c**) third, (**d**) fourth, (**e**) fifth, and (**f**) sixth cycles of reflow.

**Figure 5 materials-16-04360-f005:**
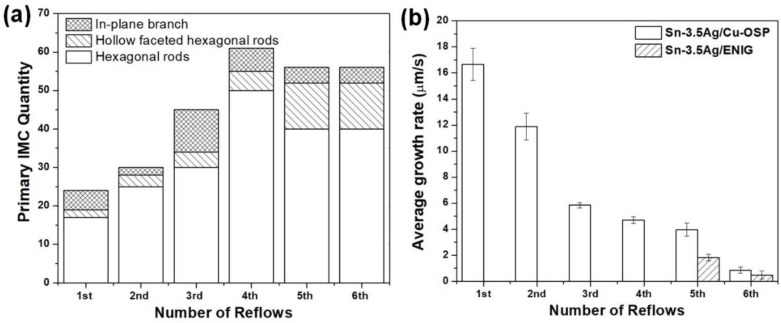
Quantification of primary intermetallics nucleation in Figure 3 for Sn-3.5Ag/Cu-OSP solder joint: (**a**) Cu_6_Sn_5_ primary intermetallics nucleation quantity versus the number of reflows and (**b**) kinetic growth rate of primary Cu_6_Sn_5_ for Sn-3.5Ag/Cu-OSP and (Cu, Ni)_6_Sn_5_ for Sn-3.5Ag/ENIG solder joints.

**Figure 6 materials-16-04360-f006:**
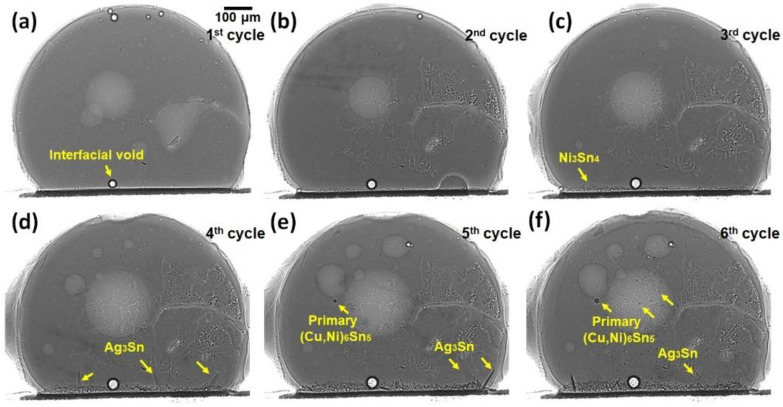
Synchrotron radiation real-time images observations of the microstructural development occurring during the multiple reflows of Sn-3.5Ag on ENIG surface finish: (**a**) first, (**b**) second, (**c**) third, (**d**) fourth, (**e**) fifth, and (**f**) sixth cycles of reflow.

**Figure 7 materials-16-04360-f007:**
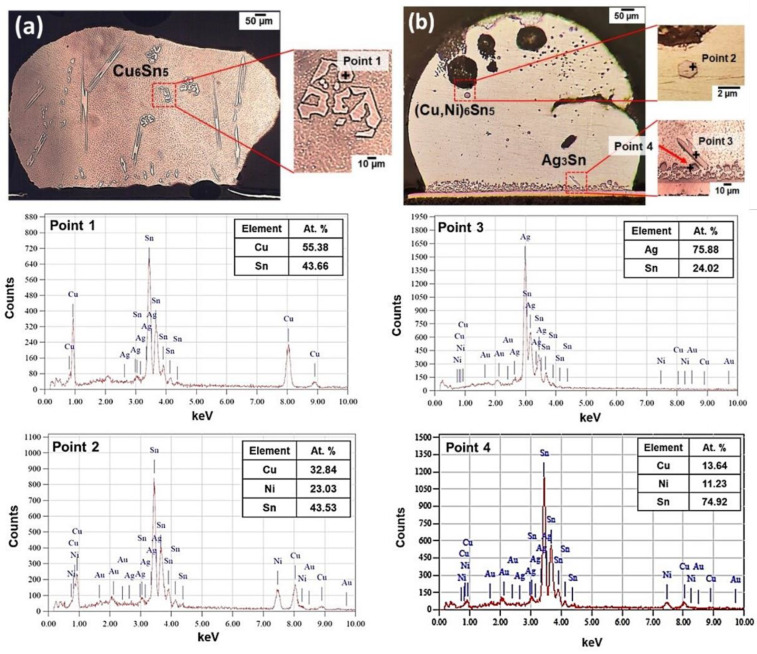
Cross-sectional image of (**a**) Sn-3.5Ag/Cu-OSP with EDS analysis of primary Cu_6_Sn_5_ intermetallics and (**b**) Sn-3.5Ag/ENIG with an EDS analysis of primary (Cu, Ni)_6_Sn_5_, Ag_3_Sn intermetallics, and (Cu, Ni)_3_Sn_4_ interfacial IMC layer. Note that both solder joint cross-sectional images were similar samples used to perform the synchrotron imaging after the sixth reflow cycle.

**Figure 8 materials-16-04360-f008:**
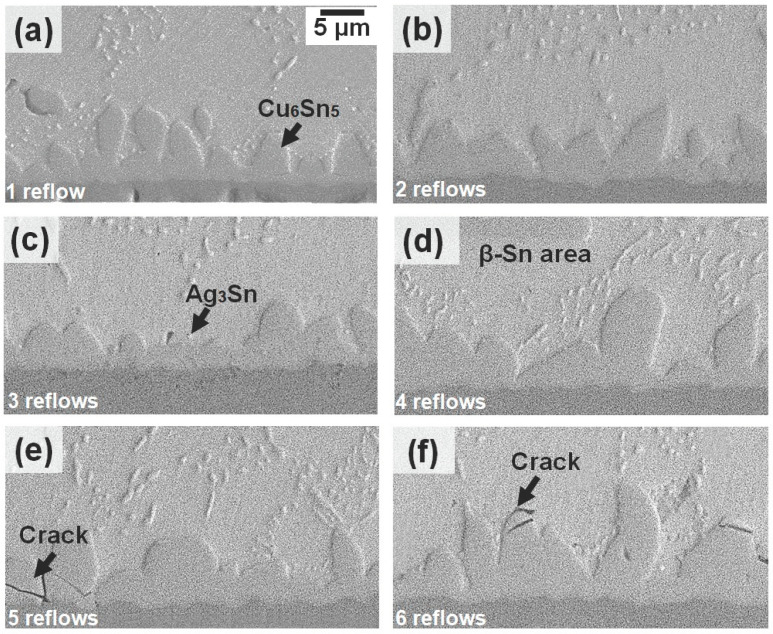
Interfacial IMC layer morphology for Sn-3.5Ag/Cu-OSP joint at different reflow cycles: (**a**) first reflow, (**b**) second reflow, (**c**) third reflow, (**d**) fourth reflow, (**e**) fifth reflow, and (**f**) sixth reflow.

**Figure 9 materials-16-04360-f009:**
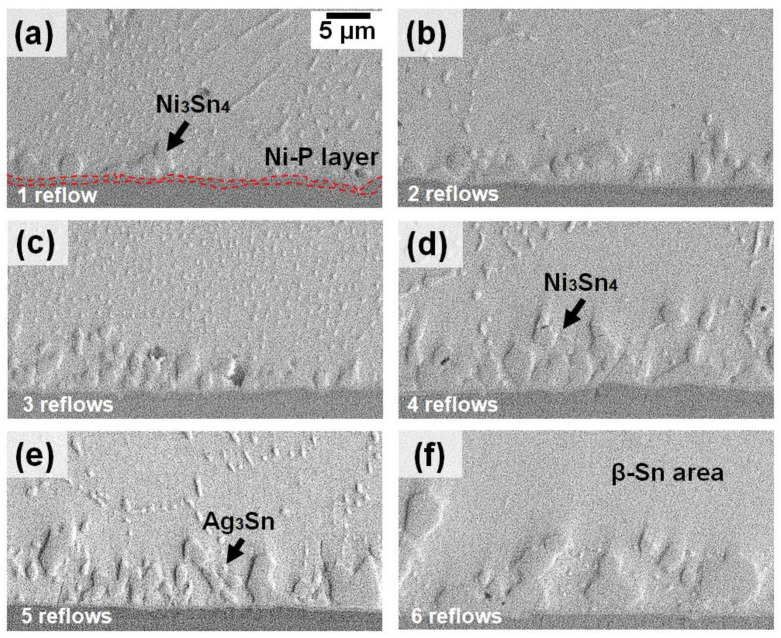
Interfacial IMC layer morphology for Sn-3.5Ag/ENIG joint at different reflow cycles: (**a**) first reflow, (**b**) second reflow, (**c**) third reflow, (**d**) fourth reflow, (**e**) fifth reflow, and (**f**) sixth reflow.

**Figure 10 materials-16-04360-f010:**
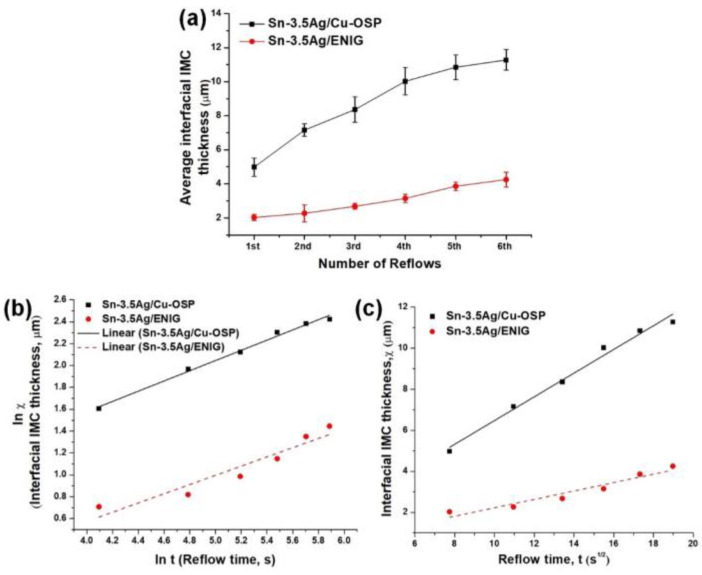
(**a**) The average interfacial thickness of Sn-3.5Ag/Cu-OSP and Sn-3.5Ag/ENIG solder joint versus the number of reflows, (**b**) ln plot of the growth of average interfacial layer versus reflow time, and (**c**) the average interfacial layer versus square root of reflow time.

**Figure 11 materials-16-04360-f011:**
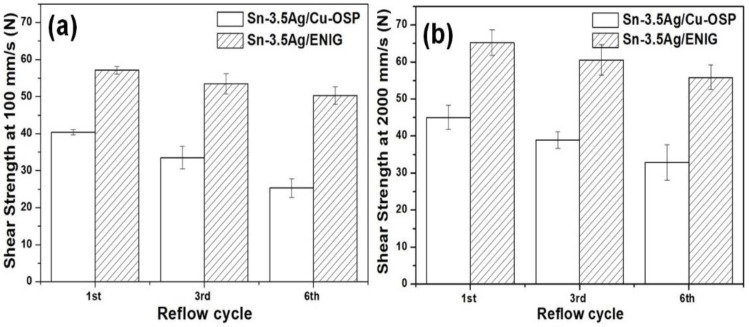
The shear strength of Sn-3.5Ag/Cu-OSP and Sn-3.5Ag/ENIG solder joint at shear speeds of (**a**) 100 mm/s and (**b**) 2000 mm/s after the first, third, and sixth cycle of reflows.

**Figure 12 materials-16-04360-f012:**
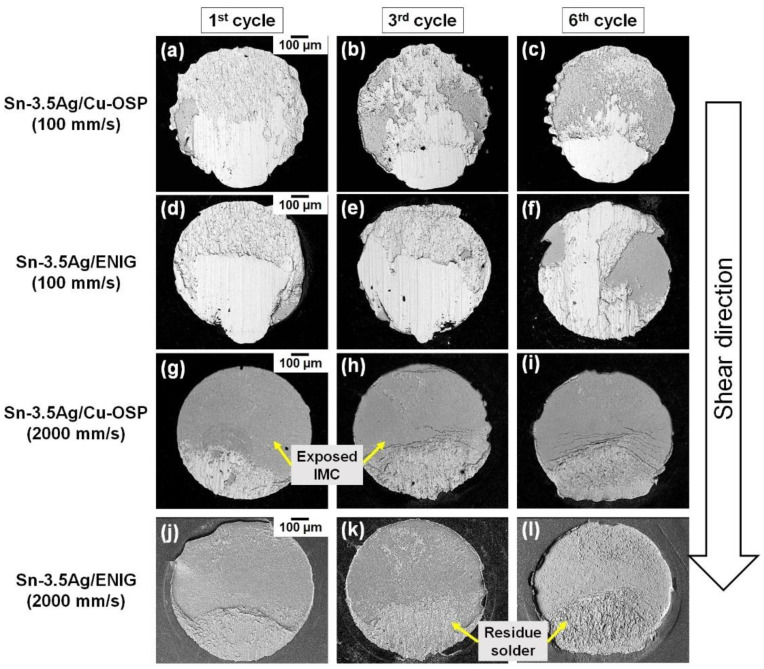
Fracture surface analysis after shear test of Sn-3.5Ag/Cu-OSP (**a**–**c**) at 100 mm/s, (**g**–**i**) at 2000 mm/s, and (**d**–**f**) for Sn-3.5Ag/ENIG at 100 mm/s, (**j**–**l**) at 2000 mm/s shear speeds.

**Figure 13 materials-16-04360-f013:**
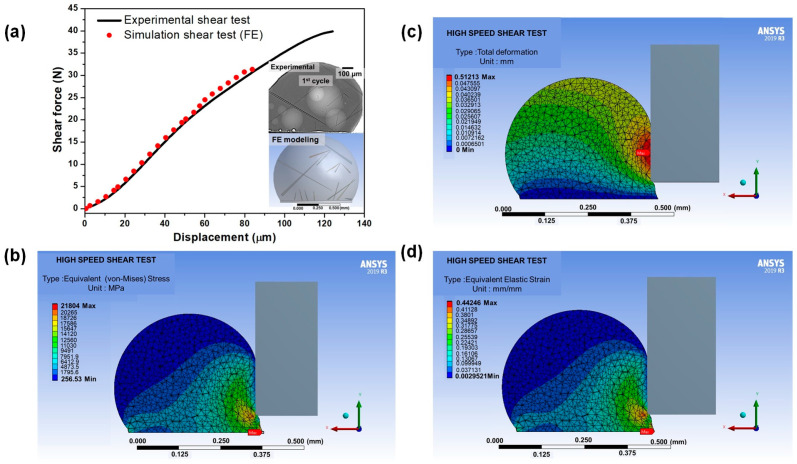
(**a**) Validation of shear force–displacement curves from the experimental and simulation, the FE contour of (**b**) von Mises stress distribution, (**c**) total deformation, and (**d**) equivalent elastic strain for Sn-3.5Ag in as-reflowed condition.

**Figure 14 materials-16-04360-f014:**
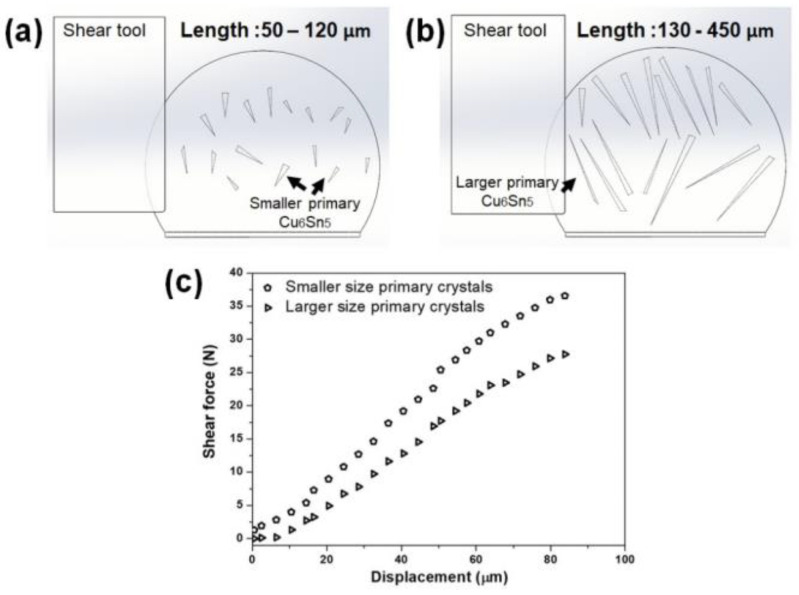
Schematic diagram of 3D finite element model with different sizes of primary Cu_6_Sn_5_ intermetallics: (**a**) smaller with the intermetallics length ranging between 50 and 120 µm, (**b**) larger with the intermetallics length ranging between 130 and 450 µm, and (**c**) shear force–displacement curves.

**Figure 15 materials-16-04360-f015:**
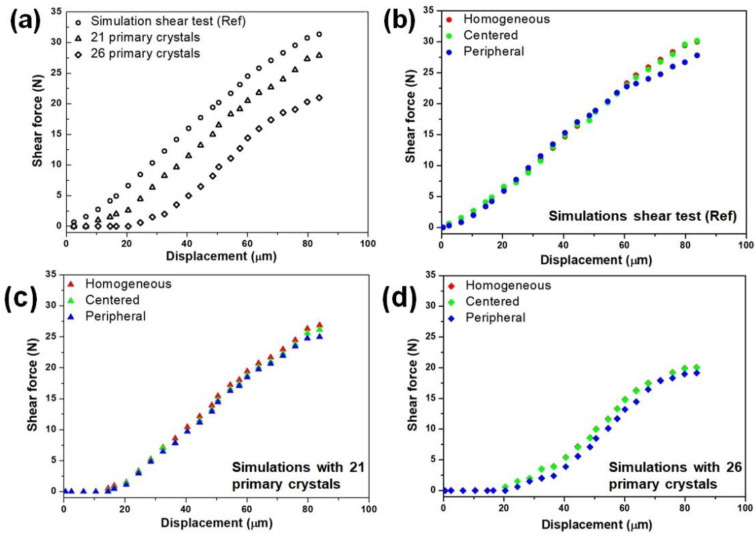
Shear force–displacement curves for (**a**) different numbers of primary and the different distributions of primary (as illustrated in Figure 3), where (**b**) for simulation shear test with 16 (**c**) 21 and (**d**) 26 primary crystals.

**Table 1 materials-16-04360-t001:** Linear elastic material properties.

Materials	Young Modulus, E (MPa)	Density, 𝜌 (g/cm^3^)	Poisson’s Ratio, *v*	Refs.
Sn-3.5Ag solder	49,800	7.5	0.4	[9]
Cu_6_Sn_5_ Intermetallics	110,000	-	0.3	[10]
Shear tool	Rigid	7.85	-	[11]

**Table 2 materials-16-04360-t002:** Input parameters for steady state creep analysis.

Materials	C_1_ (1/s)	C_2_ (1/kPa)	C_3_	C_4_ (K)	Ref.
Sn-3.5Ag solder	9.00 × 10^5^	4.5 × 10^−4^	0.182	8690	[12]

## Data Availability

The data presented in this study are available on request from the corresponding author.

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
