# Peer review of "Effects of Multiple Reflow on the Formation of Primary Crystals in Sn-3.5Ag and Solder Joint Strength: Experimental and Finite Element Analysis"

_materials, 2023, doi:10.3390/ma16124360_

Round 1

Reviewer 1 Report

The work, which uses advanced techniques is quite interesting. However the quality of the paper needs to be improved.

* Extensive editing of the english language and style are required, since many sentences are hardly understandable

* All the acronyms should be explained in the text, rather than in the abstract

* the caption of Figure 1 should be revised, since there is no part labelled e

* the spatial resolution of tomographic images (size of a Voxel) should be specified

* on page 4, line 145, when mentioning "the shear height", a reference to Fig. 1d should be made, and this term should be defined on this figure.

* it is difficult for the reader to recognize the 3 morphologies of IMC particles on the tomographic images. So, why not show 3D tomographic zoomed images (or drawings) of each type ?

* the quality of Fig. 7 should be improved

* the way the random distribution of intermetallic particles was generated in the FE model is not clear: only their size and position is mentionned. What about their orientation in space  and aspect ratio? Also a size distribution, rather than just a mean value was apparently used. Please clarify

* Since inertia effects have probably not been included in the FE simulations, it is useless to mention the densities in Table 1. Was a finite strain  or large strain formulation used ? Are Von Mises stress and equivalent strain (by the way, not "equivalent elastic strain", as written on page 15, line 471,and in the caption of Fig. 13, but "equivalent plastic strain"),  relevant for tiny solder balls that probably contain very few, higly anisotropic crystals ? Do you have any information on that point ?

* The way the simulations of Fig. 14 and 16 were done is not clear: the size or number of the Cu6Sn5 particles was changed, but was the total volume fraction kept unchanged or not? The comments of Fig. 14c (page 15, lines 500-502) and 15a (lines 507-508) should be revised. Since damage was not simulated, the brittleness of CuSn5 crystals (not described by a fracture strain in the model) has nothing to do with the lower computed shear flow stress. Since this particles have a nearly twice higher elastic stiffness than the matrix, one would expect the shear flow stress to increase with their volume fraction. That is why it is important to specify it, and to use it to provide a convincing explanation of Fig. 14c and 15a

Author Response

Ref Paper: materials-2337550

Title: "Effects of Multiple Reflow on the Formation of Primary Crystals in Sn-3.5Ag and Solder Joint Strength: Experimental and Finite Element Analysis"

Dear Editor,                                                                                                      

Please find the reviewer's comments and corrections been made by authors as below:

Reviewer: 1

No

Comment

Correction

Page/Line

(As highlighted in manuscript)

1.

Extensive editing of the english language and style are required, since many sentences are hardly understandable

Thank you for the comment. We have submitted the paper for English proofread

2.

All the acronyms should be explained in the text, rather than in the abstract

Thank you for the comments. All the acronyms have been explained in the text and highlighted in the manuscript.

2/51

2/72

2/86-87

3.

the caption of Figure 1 should be revised, since there is no part labelled e

Thank you for the comments. The caption in Figure 1 has been revised as in the manuscript.

3/109-112

4.

the spatial resolution of tomographic images (size of a Voxel) should be specified

Thank you for the comments. The spatial resolution for the images has been added and highlighted in the manuscript.

3/126

5.

on page 4, line 145, when mentioning "the shear height", a reference to Fig. 1d should be made, and this term should be defined on this figure.

Thank you for the comments. Figure 1 has been revised and "the shear height" has been included in the figure

3/108

6.

it is difficult for the reader to recognize the 3 morphologies of IMC particles on the tomographic images. So, why not show 3D tomographic zoomed images (or drawings) of each type ?

Thank you for the suggestion. The 3D image is the constructed image from simulation. We believe that the images will be the same as the morphology is similar and only the size and distribution were simulated. The clear images of X-ray synchrotron microstructure is presented in figure 4. 

7.

the quality of Fig. 7 should be improved

Thank you for the comments. The quality for Figure 7 has been improved.

9/304

8.

the way the random distribution of intermetallic particles was generated in the FE model is not clear: only their size and position is mentionned. What about their orientation in space  and aspect ratio? Also a size distribution, rather than just a mean value was apparently used. Please clarify

Thank you for the suggestion. In this study the simulation is only focused on the effect of size and 3 different distributions where other effects will be focused on our next study.

9.

Since inertia effects have probably not been included in the FE simulations, it is useless to mention the densities in Table 1. Was a finite strain  or large strain formulation used ? Are Von Mises stress and equivalent strain (by the way, not "equivalent elastic strain", as written on page 15, line 471,and in the caption of Fig. 13, but "equivalent plastic strain"),  relevant for tiny solder balls that probably contain very few, higly anisotropic crystals ? Do you have any information on that point ?

Thank you for your comments. We agree with the reviewer that the densities might not be useful when there is no consideration of inertia effects. However, the densities of the materials are required in the FEA simulation and to be used as a reference by the software.

In the simulation, the large strain was enabled to accurately represent large deformations and nonlinear material behavior, which are relevant in solder joint analysis where significant deformations can occur during the shear test.

In the simulation, the solder ball is defined as elastic-viscoplastic. Therefore, the software displayed the equivalent elastic strain to describe the strain distribution on the solder ball, as shown in Figure 13.

In terms of the relevance of Von Mises stress and equivalent plastic strain for tiny solder balls containing few highly anisotropic crystals, we acknowledge that the behavior of such small-scale structures may deviate from macroscopic assumptions. The applicability of these measures to tiny solder balls depends on the accuracy of the constitutive models and material properties employed in the analysis. However, in the absence of specific information on the crystal structure and anisotropy of the solder balls in the current study, the use of Von Mises stress and equivalent plastic strain provides a reasonable approximation for evaluating the solder joints' mechanical behavior and failure criteria.

10.

The way the simulations of Fig. 14 and 16 were done is not clear: the size or number of the Cu6Sn5 particles was changed, but was the total volume fraction kept unchanged or not? The comments of Fig. 14c (page 15, lines 500-502) and 15a (lines 507-508) should be revised. Since damage was not simulated, the brittleness of CuSn5 crystals (not described by a fracture strain in the model) has nothing to do with the lower computed shear flow stress. Since this particles have a nearly twice higher elastic stiffness than the matrix, one would expect the shear flow stress to increase with their volume fraction. That is why it is important to specify it, and to use it to provide a convincing explanation of Fig. 14c and 15a

Thank you for your comments. The total volume fraction was changed relatively when the Cu6Sn5 particles changed.

The comments of Fig. 14c and 15a have been revised as suggested.

Damage was not included in our model; the brittleness of Cu6Sn5 crystals is not explicitly accounted for by a fracture strain. Regarding the lower computed shear flow stress, we acknowledge the expectation that the shear flow stress should increase with the volume fraction of Cu6Sn5 particles due to their higher elastic stiffness compared to the matrix. The volume fraction information has been included in the revised manuscript.

We thank you for the valuable suggestions made by the reviewers to improve our paper. All concerns and suggestions have been addressed in the above feedback. We would like to thank the editor in advance for considering our work.

Yours sincerely,

Dr Mohd Arif Anuar Mohd Salleh

Nihon Superior Electronic Material Research Lab, Center of Excellence Geopolymer & Green Technology (CeGeoGTech), Faculty of Chemical Engineering & Technology, Universiti Malaysia Perlis (UniMAP), Taman Muhibbah, 02600, Arau, Perlis, Malaysia.

Reviewer 2 Report

This paper is well organized, and it provides a good comparison between Sn-3.5Ag solder/ OSP finish and Sn-3.5Ag solder/ Enig finish in terms of the solder mechanical reliability with a complete explanation. Just a few comments here.

1. Line 108, either (d) or (e) schematic in Figure 1 is missing.

2. In Section 2.5, the author creates the solder geometry by mimicking the solder joint cross-section image which may not be very accurate given that we would not be able to know the cross-section surface location. For the author's info, it can give you a more realistic solder joint geometry using Surface Evolver to simulate the solder joint shape. Please refer to [K. Pan, J. H. Ha, H. Wang, J. Xu and S. Park, "An Analysis of Solder Joint Formation and Self-Alignment of Chip Capacitors," in IEEE Transactions on Components, Packaging and Manufacturing Technology, vol. 11, no. 1, pp. 161-168, Jan. 2021, doi: 10.1109/TCPMT.2020.3034211. ]

3. How is the brittleness of the IMC captured in the FEA model? 

Author Response

Ref Paper: materials-2337550

Title: “Effects of Multiple Reflow on the Formation of Primary Crystals in Sn-3.5Ag and Solder Joint Strength: Experimental and Finite Element Analysis”

Dear Editor,                                                                                                      

Please find the reviewer’s comments and corrections been made by authors as below:

Reviewer: 2

No

Comment

Correction

Page/Line

(As highlighted in manuscript)

1.

Line 108, either (d) or (e) schematic in Figure 1 is missing.

Thank you for your comments. The caption for Figure 1 has been corrected as in manuscript

3/109-112

2.

In Section 2.5, the author creates the solder geometry by mimicking the solder joint cross-section image which may not be very accurate given that we would not be able to know the cross-section surface location. For the author's info, it can give you a more realistic solder joint geometry using Surface Evolver to simulate the solder joint shape. Please refer to [K. Pan, J. H. Ha, H. Wang, J. Xu and S. Park, "An Analysis of Solder Joint Formation and Self-Alignment of Chip Capacitors," in IEEE Transactions on Components, Packaging and Manufacturing Technology, vol. 11, no. 1, pp. 161-168, Jan. 2021, doi: 10.1109/TCPMT.2020.3034211. ]

Thank you for your comments. We agree with the reviewer that the surface location might not be very accurate compared to using Surface Evolver in terms of the solder geometrical and physical constraints. In the current FEA analysis, the solder joint geometric model was constructed based on the actual dimension from the experiment using Solidworks software, as in Figure 2 (a), and consists of the solder ball and primary intermetallics. We thank the reviewer for providing valuable information for our consideration in future works.

3.

How is the brittleness of the IMC captured in the FEA model?

Thank you for your comments. The brittleness of IMCs can be accounted for by incorporating appropriate material properties, such as Young's modulus into the FEA model.

We thank you for the valuable suggestions made by the reviewers to improve our paper. All concerns and suggestions have been addressed in the above feedback. We would like to thank the editor in advance for considering our work.

Yours sincerely,

Dr Mohd Arif Anuar Mohd Salleh

Nihon Superior Electronic Material Research Lab, Center of Excellence Geopolymer & Green Technology (CeGeoGTech), Faculty of Chemical Engineering & Technology, Universiti Malaysia Perlis (UniMAP), Taman Muhibbah, 02600, Arau, Perlis, Malaysia.

Reviewer 3 Report

The manuscript, “Effects of Multiple Reflow on the Formation of Primary Crystals in Sn-3.5Ag and Solder Joint Strength: Experimental and Finite Element Analysis” by, S. Amli et al. describes effects of multiple reflow process on a Sn-3.5Ag solder alloy based on experimental and theoretical results.  This manuscript is suitable for publication in Materials.  Comments and questions to authors are in the followings;

- This manuscript explains the relationship between the formation of primary crystals and solder joint strength.  Discussions are based on in-situ approaches with heating and cooling cycles.  In-situ synchrotron imaging measurements were performed with 0.33 °C/s of ramp rate.  However, this ramp rate looks much slower than the one in real life.  I wonder if this alters the formation of primary crystals in the soldering material.

- In this study, EDS was used to identify the phase of materials formed during reflow processes.  For example, the authors confirmed the formation of Cu6Sn5 and Ag3Sn based on the EDS results.  However, this might cause an error because EDS shows only distribution of elements.  Elements can be in metallic phase not in alloy.  I am wondering how authors confirmed the phase of materials using the EDS results?

- In Fig. 9, authors describes the presence of Ni-P layer in the early stage.  Does it stay over the reflow processes or does it disappears?

- It might be better to combine Fig. 4 and Fig. 6.  This should help readers to compare synchrotron imaging results on Sn-3.5Ag on Cu-OSP and ENIG.

- I can’t read numbers in the legend in Fig. 13.

- There is a missing parenthesis in Eq. (1).

Author Response

Ref Paper: materials-2337550

Title: “Effects of Multiple Reflow on the Formation of Primary Crystals in Sn-3.5Ag and Solder Joint Strength: Experimental and Finite Element Analysis”

Dear Editor,                                                                                                      

Please find the reviewer’s comments and corrections been made by authors as below:

Reviewer: 3

No

Comment

Correction

Page/Line

(As highlighted in manuscript)

1.

This manuscript explains the relationship between the formation of primary crystals and solder joint strength.  Discussions are based on in-situ approaches with heating and cooling cycles.  In-situ synchrotron imaging measurements were performed with 0.33 °C/s of ramp rate.  However, this ramp rate looks much slower than the one in real life.  I wonder if this alters the formation of primary crystals in the soldering material.

Thank you for the comments. Yes we agree that the different cooling rates will dictate the shapes of primary crystals in the soldering material. This observations was supported by the research publish by:

-Xian, J. W., Belyakov, S. A., Ollivier, M., Nogita, K., Yasuda, H., & Gourlay, C. M. (2017). Cu6Sn5 crystal growth mechanisms during solidification of electronic interconnections. Acta Materialia, 126, 540–551. https://doi.org/10.1016/j.actamat.2016.12.043

However, the cooling rate used in this study also falls in the range of a usable cooling rate in lead-free soldering. This is supported by several publications.

2.

In this study, EDS was used to identify the phase of materials formed during reflow processes.  For example, the authors confirmed the formation of Cu6Sn5 and Ag3Sn based on the EDS results.  However, this might cause an error because EDS shows only distribution of elements.  Elements can be in metallic phase not in alloy.  I am wondering how authors confirmed the phase of materials using the EDS results?

Thank you for your comments. We agree that the EDS could show the distribution of elements. However, the data obtained from EDS analysis showed atomic % and weight % of elements, where it corresponds to the typical phases formed in the Sn-3.5Ag solder on Cu substrate which consists of Cu6Sn5 and Ag3Sn phases. This is also been referred to the ternary phase diagram of Sn-Ag-Cu.

3.

In Fig. 9, authors describes the presence of Ni-P layer in the early stage.  Does it stay over the reflow processes or does it disappears?

Thank you for the comments. The Ni-P layer will stay over the reflow processes. This observation was also being supported by:

1.       Lin, Y. C., Wang, K. J., & Duh, J. G. (2010). Detailed phase evolution of a phosphorous-rich layer and formation of the Ni-Sn-P compound in Sn-Ag-Cu/electroplated Ni-P solder joints. Journal of Electronic Materials, 39(3), 283–294. https://doi.org/10.1007/s11664-009-1014-x.

2.       Wojewoda-Budka, J., Huber, Z., Litynska-Dobrzynska, L., Sobczak, N., & Zieba, P. (2013). Microstructure and chemistry of the SAC/ENIG interconnections. Materials Chemistry and Physics, 139(1), 276–280. https://doi.org/10.1016/j.matchemphys.2013.01.035

4.

It might be better to combine Fig. 4 and Fig. 6.  This should help readers to compare synchrotron imaging results on Sn-3.5Ag on Cu-OSP and ENIG

Thank you for your suggestion, we have separated the images as we would want the readers to observe the images clearer. The explanation of the results observed were separated according to the images and subsequently were compared in the discussion.

5.

I can’t read numbers in the legend in Fig. 13.

Thank you for the comments. The quality of the figure has been improved as in the manuscript.

15/488

6.

There is a missing parenthesis in Eq. (1).

Thank you for the comments. Eq.1 has been corrected as in manuscript.

4/168

We thank you for the valuable suggestions made by the reviewers to improve our paper. All concerns and suggestions have been addressed in the above feedback. We would like to thank the editor in advance for considering our work.

Yours sincerely,

Dr Mohd Arif Anuar Mohd Salleh

Nihon Superior Electronic Material Research Lab, Center of Excellence Geopolymer & Green Technology (CeGeoGTech), Faculty of Chemical Engineering & Technology, Universiti Malaysia Perlis (UniMAP), Taman Muhibbah, 02600, Arau, Perlis, Malaysia.
